# Prediction of Mechanical Properties of Rubberized Concrete Incorporating Fly Ash and Nano Silica by Artificial Neural Network Technique

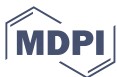

**Musa Adamu** [1,*] **, Andaç Batur Çolak** [2] **, Yasser E. Ibrahim** [1,*] **, Sadi I. Haruna** [3] **and Mukhtar Fatihu Hamza** [4]

1    Engineering Management Department, College of Engineering, Prince Sultan University, Riyadh 11586, Saudi Arabia
2    Information Technologies Application and Research Center, Istanbul Commerce University, Istanbul 34445, Turkey
3    Department of Civil Engineering, Bayero University Kano, Kano 700223, Nigeria
4    Department of Mechanical Engineering, College of Engineering in Alkharj, Prince Sattam bin Abdulaziz University, Alkharj 11942, Saudi Arabia
*    Correspondence: madamu@psu.edu.sa (M.A.); ymansour@psu.edu.sa (Y.E.I.)

**Abstract:** The use of enormous amounts of material is required for production. Due to the current emphasis on the environment and sustainability of materials, waste products and by-products, including silica fume and fly ash (FA), are incorporated into concrete as a substitute partially for cement. Additionally, concrete fine aggregate has indeed been largely replaced by waste materials like crumb rubber (CR), thus it reduces the mechanical properties but improved some other properties of the concrete. To decrease the detrimental effects of the CR, concrete is therefore enhanced with nanomaterials such nano silica (NS). The concrete mechanical properties are essential for the designing and constRuction of concrete structures. Concrete with several variables can have its mechanical characteristics predicted by an artificial neural network (ANN) technique. Using ANN approaches, this paper predict the mechanical characteristics of concrete constructed with FA as a partial substitute for cement, CR as a partial replacement for fine aggregate, and NS as an addition. Using an artificial neural network (ANN) technique, the mechanical characteristics investigated comprise splitting tensile strength (Fs), compressive strength (Fc), modulus of elasticity (Ec) and flexural strength (Ff). The ANN model was used to train and test the dataset obtained from the experimental program. Fc, Fs, $F_f$ and Ec were predicted from added admixtures such as CR, NS, FA and curing age (P). The modelling result indicated that ANN predicted the strength with high accuracy. The proportional deviation mean (MoD) values calculated for $F_c$, $F_s$, $F_f$ and $E_c$ values were $-0.28\%$, $0.14\%$, $0.87\%$ and $1.17\%$, respectively, which are closed to zero line. The resulting ANN model's mean square error (MSE) values and coefficient of determination ($R^2$) are $6.45 \times 10^{-2}$ and $0.99496$, respectively.

**Keywords:** crumb rubber; fly ash; nano silica; mechanical characteristics; artificial neural network

**MSC:** 68T07



## 1. Introduction

Numerous works have been published in the literature to better understand how well concrete performs mechanically, which is one of the most commonly used artificial materials in the construction industry. The conceptualization and construction of civil engineering structures largely depend on the laboratory measurements of the concrete's strength properties, which are subjected to environmental condition such as temperature, humidity, age and concrete compositions etc. [1–3]. In addition, various admixtures modified both fresh and hardeening properties of concrete which include crumb rubber (CR) [4,5] nanomaterials [6–9], polymer materials [10–12], fiber-reinforcement [13–17], and pozzolonic materials [18–20].

Other waste materials/recyching materials were also used for modifications of different types of concrete, for example crushed glass waste was used as partial replacement to aggregates aggregate in concrete and was redported to enhance the flexural strength of the concrete [21]; waste glass powder as partial substitute to cement decreased the slump and mechanical strengths of concrete, but when used as partial replacement to aggregates it improves the concrete's strengths [22,23]; waste lathe scrap were also found to improve the compressive strength and mechanical features of reinforced concrete beams [24]; recycled coal bottom ash as replacement to fine aggregate was found to increase the deflection of reinforced concrete beams [25]; waste marble waste as partial subsitutue to cement was found to reduce the compressive strength and crack behavior of reinforced concrete beams [26]. Therefore, mechanical properties of concrete depends mainly on its constituent material and admixtures [5,27,28]. Presently, artificial intelligent technique have been demonstrating a rubost capacity in training complex dataset of estimating purpose. For instance, Chou, Tsai [29] created an ensemble model using an ANN model, a support vector machine (SVM) for estimating the concrete's strength. The compressive strength of the existing concrete structure has being estimating through different input parameter using ANN model [30]. Chopra, Sharma [31] used genetic programming and an ANN model to analyse the concrete's compressive strength. Rebound hammer and UPV deta sets were applied, as input parameters.The employment of artificial intelligence tools in data forecasting has been widely used recently. Among the artificial intelligence technologies utilised in various technical domains is the ANN, which was inspired by the biological configuration of humans. In certain works of literature, the mechanical characteristics of cement mortars and concrete materials were modelled using ANN. Jang and Xing [32] measured the emissions of ammonia from mortar containing various fly ash types and observed a strongly relationships between the emissions and the fly ash contents, mortar size and age. Then, they used ANN models for predicting the concentration of ammonia under different conditions. The study's results showed that the genetic algorithm ANN models had the least root mean square error (RMSE) when compared to real outputs. Althoey, Akhter [33] compared the performace of different model for Marshall Mix Parameters Using Bio-inspired Genetic Programming and Deep Machine Learning techniques. The comparison analysis result indicates that ANN, ANFIS, MEP and DT-Bagging are all effective and reliable technique for the estimation of Marshall Stability and Marshall Flow. Similarly, Madenci and Özkılıç [34] explore the effect of porosity on free vibration analysis of functionally graded (FG) beams using analytical and numerical approaches. An ANN model and backpropagation technique were used by Felix, Carrazedo [35] to estimate the depth of carbonation in concretes containing fly ash. For networks with two hidden layers, the ANN model that was developed utilizing a collection of 90 data points, can create models with determination coefficient higher than 0.8. The optimized configuration was able to give the smallest RMMSE linked with the highest coefficient of determination. Based on the parametric study, it was found that fly ash and, $CO_2$ ratio, cement contents and relative humidity were the primary factor which influence the carbonation depth in fly ash-concrete. The suggested models could also be used for simulating the growth of engineering projects focused on durability as well as to anticipate the lifespan of concrete structures. In the study by Pazouki [36], three different models including, group data processing technique, optimization algorithm of ant colony and ANN supported radial based functional neural network are proposed for predicting the compressive strength of the fly ash originated from geopolymer concrete. In this study, 360 samples of fly ash-based geopolymer concrete were used to generate the data set for this investigation. An accuracy and estimation capacity of the models were evaluated with statistical formulas and the models were correlated with an experimental test result. The research revealed that all of the models could predict this mechanical characteristic of fly ash-originated from geopolymer concrete with acceptable accuracy, but that the radial-based function neural network had the best correlation when correlated with the other models. Models to predict the drying shrinkage of alkali-activated fly ash-black furnace mortars were developed by Adesanya, Aladejare [37]. In the experimental study,

several factors were taken into account, and the impact of mortars on drying shrinkage was examined. The drying shrinkage of the mortar at 28 days were estimated using multiple linear regression and ANN models. Based on the validations, the experimental results and ANN models were highly correlated. Çelik, Yildiz [38] proposed an ANN model to examine how nano silica combined with a fly ash affects rheological characteristics for cementitious mortars. Experimental research was conducted on the effects of nanosilica as an additive upon the plastic viscosities and values of yield stress for cement-based mortars with varying concentrations of volatile additives as ash mineral additives. In order to use the results of the experiments to evaluate the plastic viscosities and yield stresses of cement originated from mortars incorporating nanosilica, a feedforward backpropagation ANN model has been developed. The proposed models were stated to have very good predictive accuracy.ANN modelling is getting more popularity and has been used to solve several engineering tasks. ANN recorded many successes in civil and structural engineering application. It has been used to predicts concrete durability [39], estimating load-displacement curve of concrete [40], and concrete strength [41]. The fundamental benefit of ANNs is that no particular equation is required. The relationship between variables is automatically managed by ANN, which also is adapted according to the training dataset. Several studies have employed the application of many models, which include ensemble model, hybrid, and boosted model. However, few studies evaluated the concrete mechanical properties using a single model. Due to its extremely robust and effectiveness in solving complicated problems, the study employed ANN model to evaluate mechanical of concrete. Therefore, the aim of this study was to predict the mechanical properties of concrete containing crumb rubber, fly ash, and nano silica. The properties of the concrete predicted includes compressive and flexural strengths, and modulus of elasticity. The predicted models will help to reduce the number of experiments required and save cost and time.

## 2. Materials and Methods

### 2.1. Materials

The primary binder material was Grade 42.5R cement (Type I), which meets the standards for cement composition as forth in ASTM C150 [42]. Table 1 provides a summary of the cement's characteristics. A white dispersive powder form of commercially available nanosilica obtained from Zhengzhou Dongshen Petrochemical Technology Co., (Zhengzhou, China) in China was added to the cementitious ingredients by weight. The amorphous structure of the nanosilica made it suitable to be utilized as a pozzolanic material as well as a filler. Table 2 provides a summary of the nanosilica's characteristics. In this investigation, cement was partially substituted in high volume with low-calcium Class F fly ash. The fly ash was obtained from YTL cement Berhad, Kuala Lumpur Malaysia. The fly ash met the ASTM C618 [43] standard criteria. Table 1 provides a summary of the fly ash's characteristics. The fine aggregate utilized was natural sand. The fine aggregate's minimum size is 4.75 mm, it has a 2.65 specific gravity, 1.24% water absorption, and a 2.86 fineness modulus. Figure 1 shows the fine aggregate's particle size gradation. The particle size gradation of the fine aggregate was carried out in accordance with ASTM C136 [44]. For the coarse aggregate, natural gravels were crushed. The maximum particle size for the coarse aggregate is 19 mm, the specific gravity is 2.66, and water absorption is 0.48 percent. The particle size analysis of the coarse aggregate was done in accordance with ASTM C136 [44], and the gradation curve is presented in Figure 1.

**Table 1.** Characteristics of binder.

| Composition of Oxides | Quantity by Mass (%) | |
| --- | --- | --- |
| | Cements | Fly Ash |
| SiO$_2$ (%) | 20.76 | 57.06 |
| CaO (%) | 61.4 | 9.79 |
| Al$_2$O$_3$ (%) | 5.54 | 20.96 |
| Fe$_2$O$_3$ | 3.35 | 4.15 |
| MgO (%) | 2.48 | 0.033 |
| Na$_2$O (%) | 0.19 | 2.23 |
| K$_2$O (%) | 0.78 | 1.53 |
| TiO$_2$ (%) | - | 0.68 |
| SO$_3$ (%) | 1.49 | - |
| Loss of ignition (%) | 2.2 | 1.25 |
| Specific gravity | 3.15 | 2.4 |
| Blaine fineness (m$^2$/kg) | 325 | 290 |

**Table 2.** Characteristics of binder.

| Items | Qualities |
| --- | --- |
| Average particle size (nm) | 10–25 |
| Hydrophobicity | Strong |
| SiO$_2$ (dry base) (%) | ≥92 |
| SiO$_2$ (%) (950 °C 2 h) | ≥99.8 |
| Specific surface area (m$^2$/g) | 100 ± 25 |
| PH value | 6.5–7.5 |
| Surface density (g/mL) | ≤0.15 |
| Hear reduction (%) (105 °C 2 h) | ≤3 |
| Loss of ignition (%) (950 °C 2 h) | ≤6 |
| Dispensability (%) (%) (CCl$_4$) | ≥80 |
| Oil-absorbed value (mL/100 g) | ≥250 |
| Hydrophobicity | Strong |

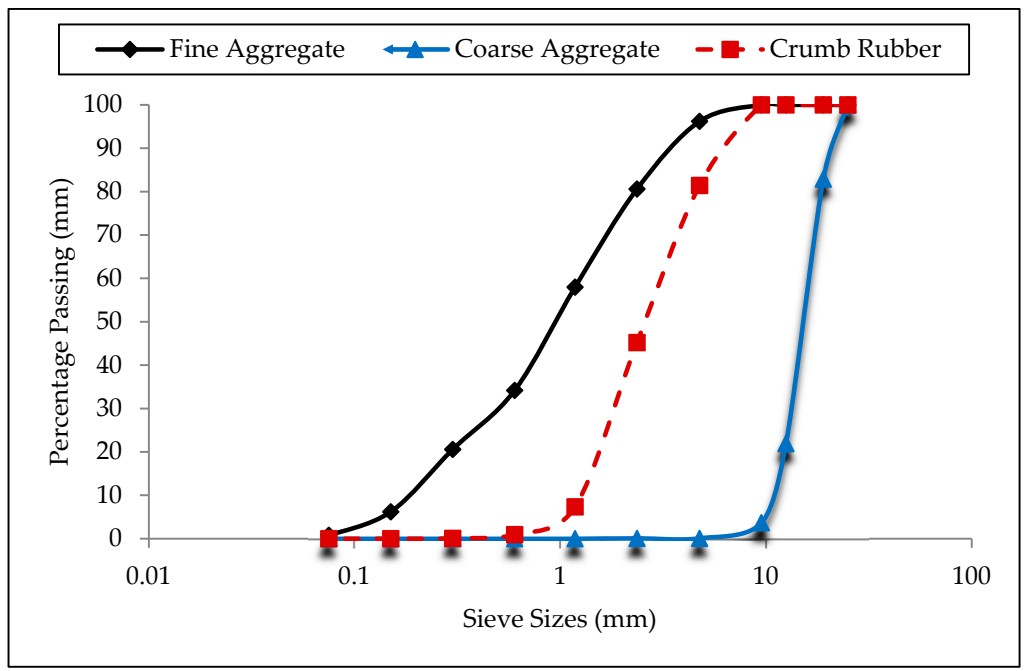

**Figure 1.** Aggregate's particle size distribution.

CR was used to partially replace fine aggregate. The CR was obtained from scrap tire after it was grinded and reduced to smaller sizes ranging in sizes between 4.75 mm to 75 μm. Before adding to the concrete, the CR was thoroughly washed using clean water to remove all dirts and impurites and then air dried for 48 h to make it completely dry. In order to get the same gradation with the fine aggregate it is replaced, three different sizes of CR were blended together. The sizes ae 3–5 mm sizes, 1–3 mm sizes, and mesh 30 (0.6 mm) sizes, in proportions of 20%, 40% and 40% correspondigly. The particle size gradation of the CR was determined using the standard procedures outlined in ASTM D5644 [45], and the gradation curveis shown in Figure 1 and was found to have similar gradation curve to the fine aggregate it partially replaced. Figure 2 presents the photos of the CR used in this study. The CR had a specific gravity of 0.93 and bulk density of 978 kg/m³ and water absorpion of 0.6%. The CR was used to replace fine aggregate in different proportions of 0%, 10%, 20% and 30% by volume fraction of the aggregate. The summary of the mass fractions of the chemical components of the CR is summarized in Table 3.

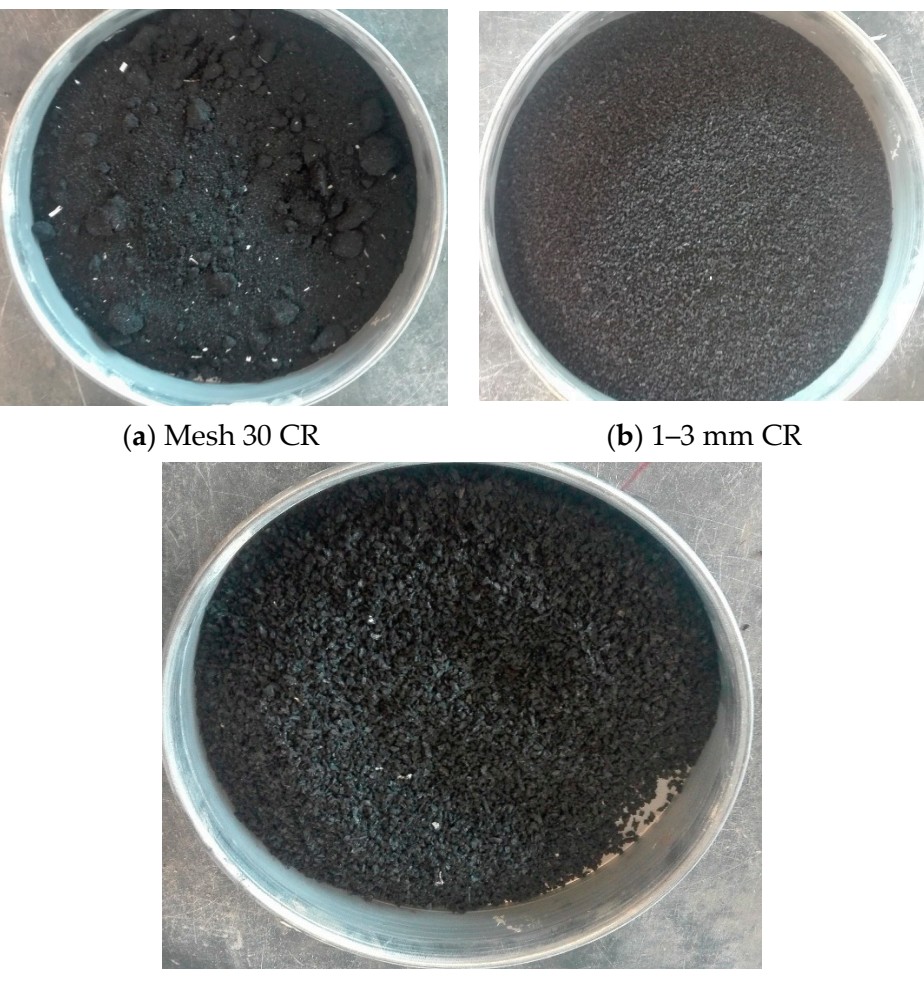

(**a**) Mesh 30 CR                                                        (**b**) 1–3 mm CR

(**c**) 3–5 mm CR

**Figure 2.** Crumb rubber used.

**Table 3.** Chemical Composition of CR. Reprinted with permission from Ref. [46] Copyright 2023 Elsevier BV.

| Chemical | C | O | Si | Zn | S | Mg | Al |
|---|---|---|---|---|---|---|---|
| Composition by Mass (%) | 87.5 | 9.24 | 0.2 | 1.77 | 1.07 | 0.14 | 0.08 |

## 2.2. Mix Proportioning

This study's concrete was roller-compacted concrete pavement (RCCP). The mixed design is completed using the geotechnical method of design (soil compaction procedure), depending on a flexural strength target of 4.8 MPa, which is equivalent to C30/37 target compressive strength grade in accordance with the guidelines ACI 211.3R [47] and CRD-C 162 [48]. 13% by weight of the dry aggregates was the cement content used. To achieve the standard gradation and requirements of ACI 211.3R [47] and CRD-C 162 [48] for RCCP, fine sand passing 75 μm sieve was used as mineral filler. The mineral filler added was 5% of the total aggregates. To make the mixtures more consistent, a high-range water reduction additive (superplasticizer) was added. Superplasticizer usage was limited to 1% of the weight of cementitious materials. In all the mixes, the percentage replacement of cement with fly ash was kept to 50%. The CR was added at different proportions of 0%, 10%, 20% and 30% by volume replacement to fine aggregate, and nanosilica was added at 0.0%, 1.0%, 2.0% and 3.0% by weight of binder materials. Table 4 displays the mix proportions and constituent ingredients for each of the mixes used for the ANN modelling.

**Table 4.** Mix Proportions.

| Mixes | Variables (%) | | | Materials Constituent (kg/m³) | | | | | | | |
|---------|---------|-----|-----|--------|---------|------|-------------------|--------|---------------------|--------|------|
| | Fly Ash | CR | NS | Cement | Fly ash | NS | Fine Aggregate | CR | Coarse Aggregate | Water | SP |
| Control | 0 | 0 | 0 | 268.69 | 0 | 0 | 1148.05 | 0 | 831.88 | 98.24 | 2.69 |
| 1 | 50 | 0 | 0 | 134.58 | 102.54 | 0 | 1150.08 | 0 | 831.88 | 96.87 | 2.37 |
| 2 | 50 | 0 | 1 | 134.58 | 102.54 | 2.37 | 1150.08 | 0 | 831.88 | 96.87 | 2.39 |
| 3 | 50 | 0 | 2 | 134.58 | 102.54 | 4.74 | 1150.08 | 0 | 831.88 | 96.87 | 2.42 |
| 4 | 50 | 0 | 3 | 134.58 | 102.54 | 7.11 | 1150.08 | 0 | 831.88 | 96.87 | 2.44 |
| 5 | 50 | 10 | 0 | 134.58 | 102.54 | 0 | 1035.07 | 115.08 | 831.88 | 96.87 | 2.37 |
| 6 | 50 | 10 | 1 | 134.58 | 102.54 | 2.37 | 1035.07 | 115.08 | 831.88 | 96.87 | 2.39 |
| 7 | 50 | 10 | 2 | 134.58 | 102.54 | 4.74 | 1035.07 | 115.08 | 831.88 | 96.87 | 2.42 |
| 8 | 50 | 10 | 3 | 134.58 | 102.54 | 7.11 | 1035.07 | 115.08 | 831.88 | 96.87 | 2.44 |
| 9 | 50 | 20 | 0 | 134.58 | 102.54 | 0 | 920.06 | 230.17 | 831.88 | 96.87 | 2.37 |
| 10 | 50 | 20 | 1 | 134.58 | 102.54 | 2.37 | 920.06 | 230.17 | 831.88 | 96.87 | 2.39 |
| 11 | 50 | 20 | 2 | 134.58 | 102.54 | 4.74 | 920.06 | 230.17 | 831.88 | 96.87 | 2.42 |
| 12 | 50 | 20 | 3 | 134.58 | 102.54 | 7.11 | 920.06 | 230.17 | 831.88 | 96.87 | 2.44 |
| 13 | 50 | 30 | 0 | 134.58 | 102.54 | 0 | 805.05 | 345.27 | 831.88 | 96.87 | 2.37 |
| 14 | 50 | 30 | 1 | 134.58 | 102.54 | 2.37 | 805.05 | 345.27 | 831.88 | 96.87 | 2.39 |
| 15 | 50 | 30 | 2 | 134.58 | 102.54 | 4.74 | 805.05 | 345.27 | 831.88 | 96.87 | 2.42 |
| 16 | 50 | 30 | 3 | 134.58 | 102.54 | 7.11 | 805.05 | 345.27 | 831.88 | 96.87 | 2.44 |

## 2.3. Testing Procedures

The concrete was weighed, batched and mixed in accordance to the procedures outlined in ASTM C192/192M [49] standards using a pan type concrete mixer as shown in Figure 3a. The freshly mixed concrete was cast into the proper moulds after a uniform mix had been achieved in accordancce with the guidelines outlined in ASTM C1435/C1435M [50]. Since RCCP is a dry, rigid mix, adequate compaction cannot be achieved using the usual compaction techniques of a vibration table or tamping rod. Therefore, A 50-Hz vibration hammer is used to compact the mixtures in the molds as shown in Figure 3b. Each layer was then crushed after being poured into the mould until a ring of mortar had formed around the plate's edge that was attached to the hammer. In compliance with the ASTM C1435/C1435M [50], this compaction was carried out. The concrete was left in the laboratory after casting for at least 24 h to settle and solidify. After samples being demolded, then stored in water till the test day.

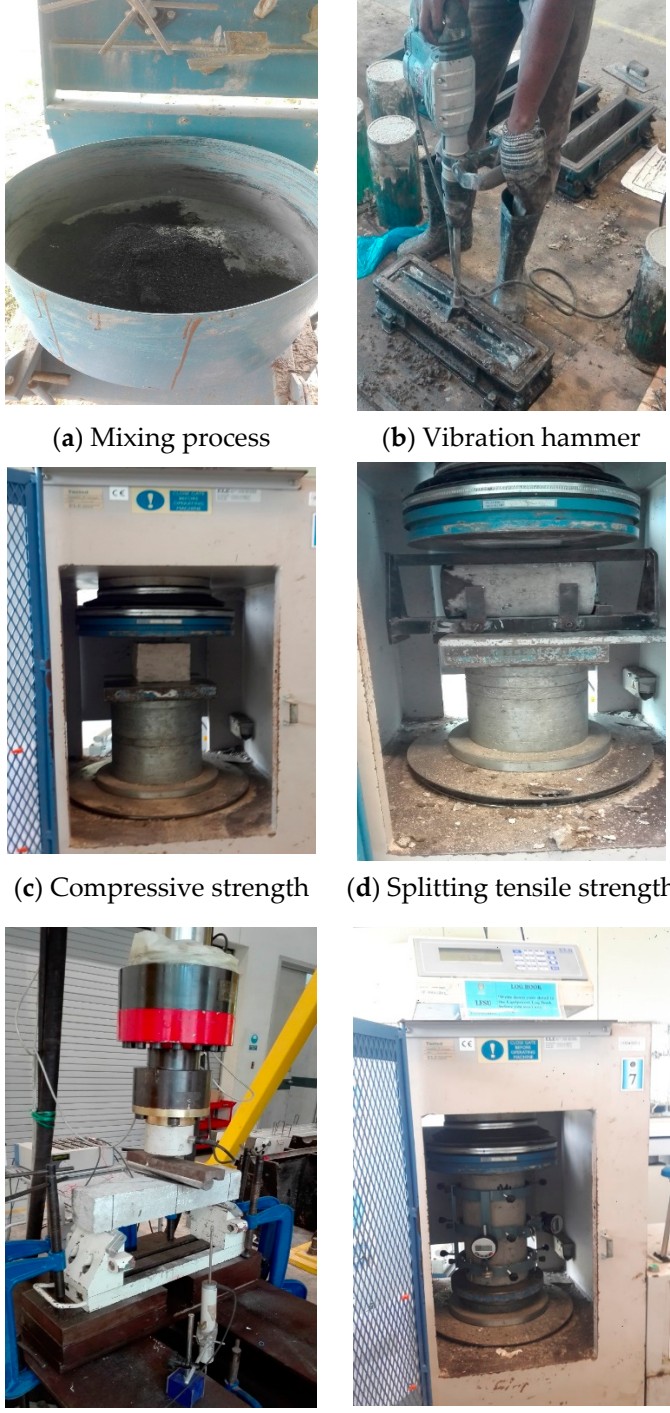

(**a**) Mixing process　　　　　　(**b**) Vibration hammer

(**c**) Compressive strength　　　　(**d**) Splitting tensile strength

(**e**) Flexural strength　　　　　　(**f**) Modulus of elasticity

**Figure 3.** Experimental setup.

According to BS EN 12390-3 [51] requirements, the compressive strength's tests for concrete was conducted using cubic specimen of (100 × 100 × 100) mm at 3, 7 and 28 days using a Universal Testing Machine (UTM) with a 2000 kN capacity as shown in Figure 3c. According to BS EN 12390-6 [52] a cylindrical sample having a diameter and height of 100 mm and 200 mm, respectively was utilized in splitting tensile test as shown in Figure 3d. A 2000 kN capacity UTM was utilized for the splitting tensile strength test. As per ASTM C293/C293M [53] specifications, beam samples measuring (100, 100, 500) mm as shown in Figure 3e were utilized for the flexural strength tests. Beam with center point load method

was used for measuring the flexural strength. A self-straining loading frame containing a 500 kN dynamic servo-controlled actuator was used for the testing. The load was applied on the samples at a constant speed of 0.1 mm/s. The modulus of elasticity test was conducted in accordance with ASTM C469/C469M [54] using a 2000 kN UTM [37], utilizing cylinder specimen having diameter and height of 150 mm 300 mm, respectively. Longitudinal and lateral compressormeters of 200 mm effective gauge lengths were mounted centrally at mid-height of each sample to capture the lateral and longitudinal strain during loading as presented in Figure 3f. The longitudinal and lateral strains were used to compute the modulus of elasticity of the samples. For ensure accuracy of each of the tests, three samples were tested for each experiment and the average value was recorded.

*2.4. ANN Modelling*

For predicting the values of Fc, Fs, Ff and Ec in relation to CR, NS, FA and P parameters, a multilayer perceptron (MLP) ANN model was created. Due to their excellent prediction performance, MLP network models, which possesses a layer design, are one of the most used models of ANN [55,56]. Fc, Fs, $F_f$ and Ec parameters are interpreted as parameters' input in the input's layer of the proposed ANN model, and CR, NS, FA and *p* values are estimated in the output's layer. Among the problems in creating MLP network is the absent of a rule for estimating neurons' number contained in the hidden layer [57]. To address this issue, performance comparisons of models comprising various neurons' number in the hidden layer led to the MLP model with 15 neurons being favored. Figures 4 and 5 show the architectural configuration and basic composition of the developed MLP network, respectively.

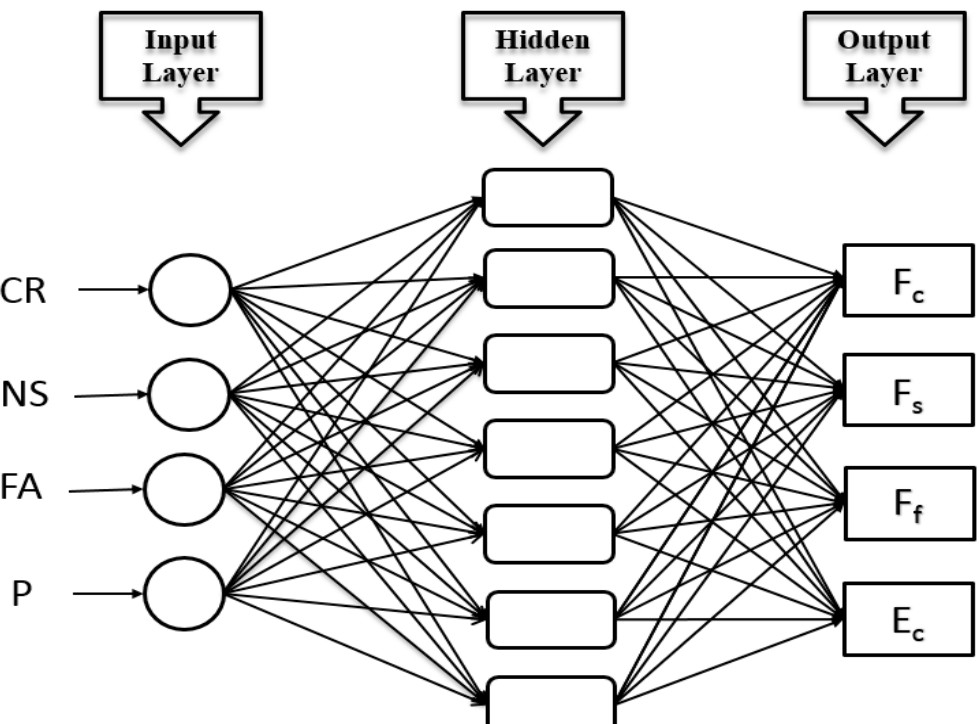

**Figure 4.** The MLP model's configuration architecture.

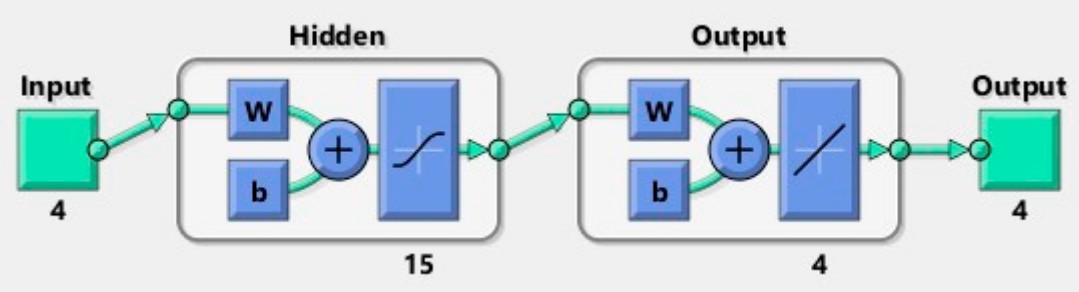

**Figure 5.** Basic structure of the developed MLP network.

It is crucial to optimise the data collection in the best possible way when creating ANN models [58]. Three major categories that are most frequently utilized in the literature constitute up the data used in the ANN model, which was created employing a total of 448 data sets [59]. 15% of the data were utilized for the model's validation, 15% for the model's testing, and 70 percent of the data were employed for the model's training. The results from the literature studies were used to choose the Levenberg-Marquards training's method that possesses a high capacity for learning, in order to train the MLP network [60]. Below is a list of the Purelin and Tan-Sig functions employed in hidden and output layer [61]:

$$f(x) = \frac{1}{1 + \exp(-x)} \tag{1}$$

$$purelin(x) = x \tag{2}$$

*2.5. Evaluation Matrices*

Equations (3)–(5) are the performance metrics employed in this study to assess the learning, training, and predicting capabilities of the developed ANN model. They are frequently used matrices for evaluating how well the proposed model performs. The formulas used to calculate the performance measures known as mean squared error (MSE), determination's coefficient (R), and a margin of deviation (MoD) are provided below [62,63]:

$$MSE = \frac{1}{N} \sum_{i=1}^{N} \left( X_{targ(i)} - X_{ANN(i)} \right)^2 \tag{3}$$

$$R = \sqrt{1 - \frac{\sum_{i=1}^{N} \left( X_{targ(i)} - X_{ANN(i)} \right)^2}{\sum_{i=1}^{N} \left( X_{targ(i)} \right)^2}} \tag{4}$$

$$MoD = \left[ \frac{X_{targ} - X_{ANN}}{X_{targ}} \right] \times 100 \ (\%) \tag{5}$$

**3. Results and Discussion**

*3.1. Optimal Choice of Input Parameters*

In order to simulate any data-driven model and provide the desired and precise results, the choice of potential input parameters is crucial. Therefore, including inappropriate parameters in artificial intelligent-based modeling reduces the developed model's performance accuracy and increases the computational difficulties [64–66]. However, inadequate input variables can lead to poor prediction accuracy. As a result, in our work, we used sensitivity analysis using Pearson's correlation to choose the critical most input parameter for predicting the fundamental mechanical characteristic for concrete incorporating admixtures. Four output parameters were consider in the modeling, and finding their idividual releationship with input parameters may be unuseful. However, compressive strength being the basic mechanical property was considered and evaluated for its correlation rela-

tionship with input parameters as depicted in the Figure 6. It can be seen that, the curing age demonstrate highest correlation value of 0.57 with compressive strength, this indicate curing age is most relevent parameter to mechanical characteristics of concrete incorporating these admixtures. Fly ash, CR and NS, however, show a negative relationship with compressive strength. Table 5 provides a summary of the dataset's statistical description.

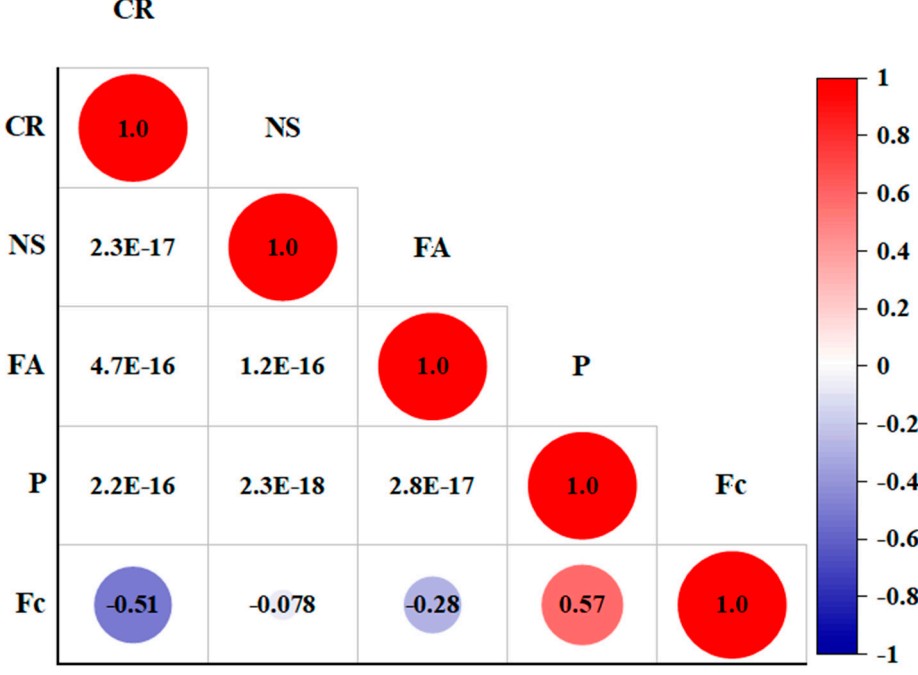

**Figure 6.** Correlation matrix using person correlation matrix.

**Table 5.** Descriptive statistic of the experimental dataset.

| Direction | Parameter | Symbols | Unit | Min | Max | Mean | SD | Kurtosis | Skewness |
|---|---|---|---|---|---|---|---|---|---|
| Inputs | Crumb rubber | CR | % | 0 | 30 | 15.00 | 11.215 | −1.365 | 0.00 |
| | Nano silica | NS | % | 0 | 3 | 1.50 | 1.121 | −1.365 | 0.00 |
| | Fly ash | FA | % | 0 | 50 | 25.00 | 25.078 | −2.025 | 0.00 |
| | Curing time | P | days | 3 | 365 | 98.60 | 137.22 | 0.017 | 1.32 |
| Output | Compressive strength | $F_c$ | MPa | 11.68 | 90.86 | 45.98 | 17.22 | −0.499 | 0.27 |
| | Splitting tensile | $F_s$ | MPa | 1.35 | 6.41 | 3.81 | 1.23 | −0.559 | 0.096 |
| | Flexural strength | $F_f$ | MPa | 2.60 | 8.89 | 5.32 | 1.32 | 0.482 | 0.707 |
| | Modulus of elasticity | $E_c$ | GPa | 5.79 | 37.78 | 19.85 | 7.53 | −0.393 | 0.440 |

The relative frequency distribution of the experimental dataset employed to predicts Fc, Fs, $F_f$ and Ec is depicted in Figure 7. The distribution plots revealed that some of the variables in the dataset follow the normal or nearly normal distribution, and some datasets do not follow the normal distribution. Most of the dataset for nano silica, crumb rubber, and fly ash have being used. The frequently used value of curing age was between 28 days and 60 days, as shown in Figure 7d. All the independent parameters follows normal distribution with thier mean value of the dataset located at the centre with highes frequency value.

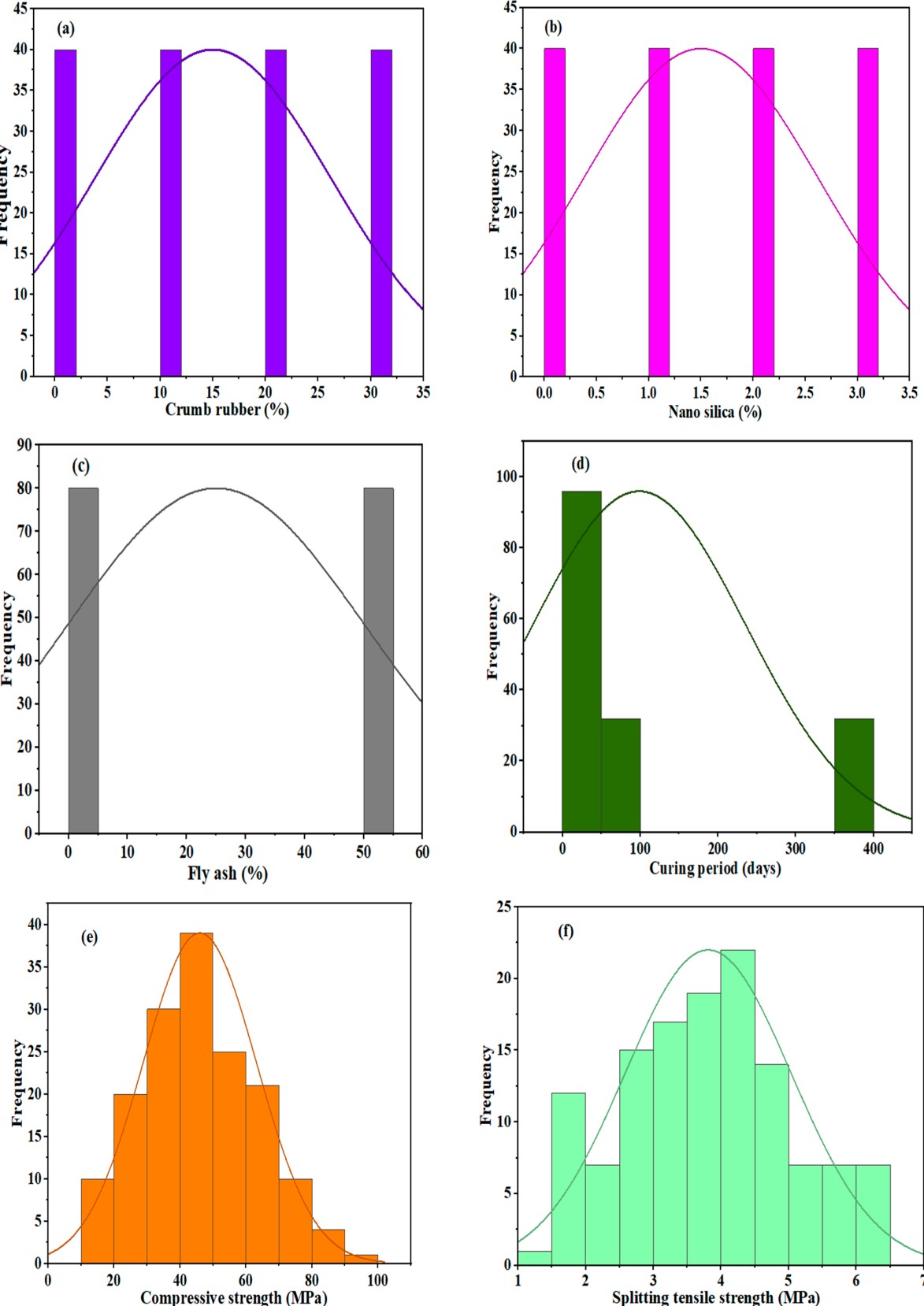

**Figure 7.** *Cont.*

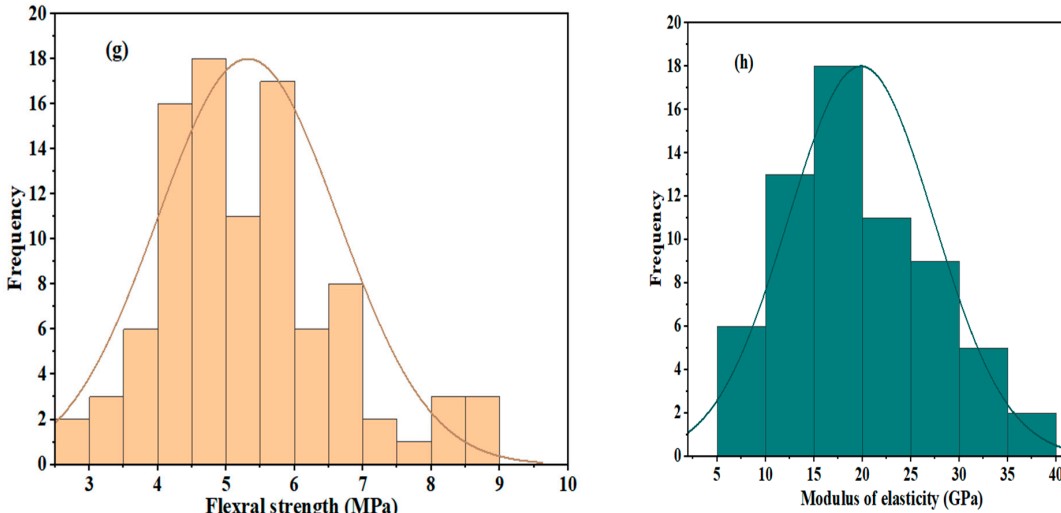

**Figure 7.** Distribution plot of the experimental dataset.

### 3.2. Modelling Results

To develop ANN model for the predictiom of mechanical properties of concrete containing admixtures, MATLAB (2021a) toolbox was used in this work. Each model was validated using the 10-fold cross-validation method [66–68]. ANN model was used to train and test the experimental datasets, including CR, fly ash, NS and curing age of the concrete, as the input parameters. On the other hand, the target parameters were calculated including, flexural strength, compressive strength, elastic modulus, and splitting tensile.

Making sure that the learning and training phases of the model are optimally completed is the first step in examining the predictive performance for ANN model. To accomplish this, it was first mandatory to look at the performance graph that the ANN model's training phase had produced. Examining the training performance graph shown in Figure 8, These MSE values are higher at the beginning of the MLP network's training step, can be shown to decrease with each passing epoch. This decline in MSE value is a sign that the deviations between the output layer's Fc, Fs, Ff and Ec values and the actual values are also declining. The 25th epoch, where the best performance was obtained for each of the three phases of the dataset, marked the end for training phase of an ANN model. The results from the performance's graph indicate that the training stage of an ANN model created for estimating the values of Fc, Fs, Ff and Ec has been completed finally.

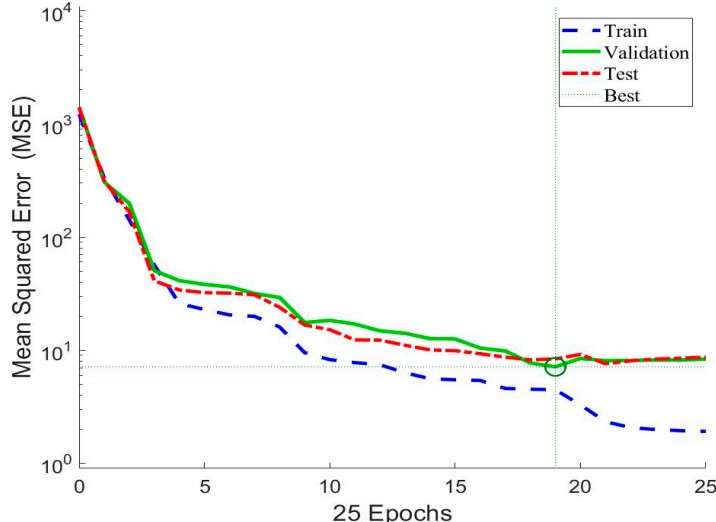

**Figure 8.** MLP model's training performance status.

The error histogram of the created ANN model is shown in Figure 9. The values of error obtained during the training stage are displayed in the error histogram. Whenever an error histogram is considered, the errors obtained across all three data groups often frequently located near the zero error line. An error histogram reveals that the errors' numerical values are also quite small. An error histogram results demonstrate that the constructed ANN model's training phase was finally completed with very few errors. The accuracy of the predictions acquired via the ANN model should be examined once the training phase has been validated Bulleted lists look like this:

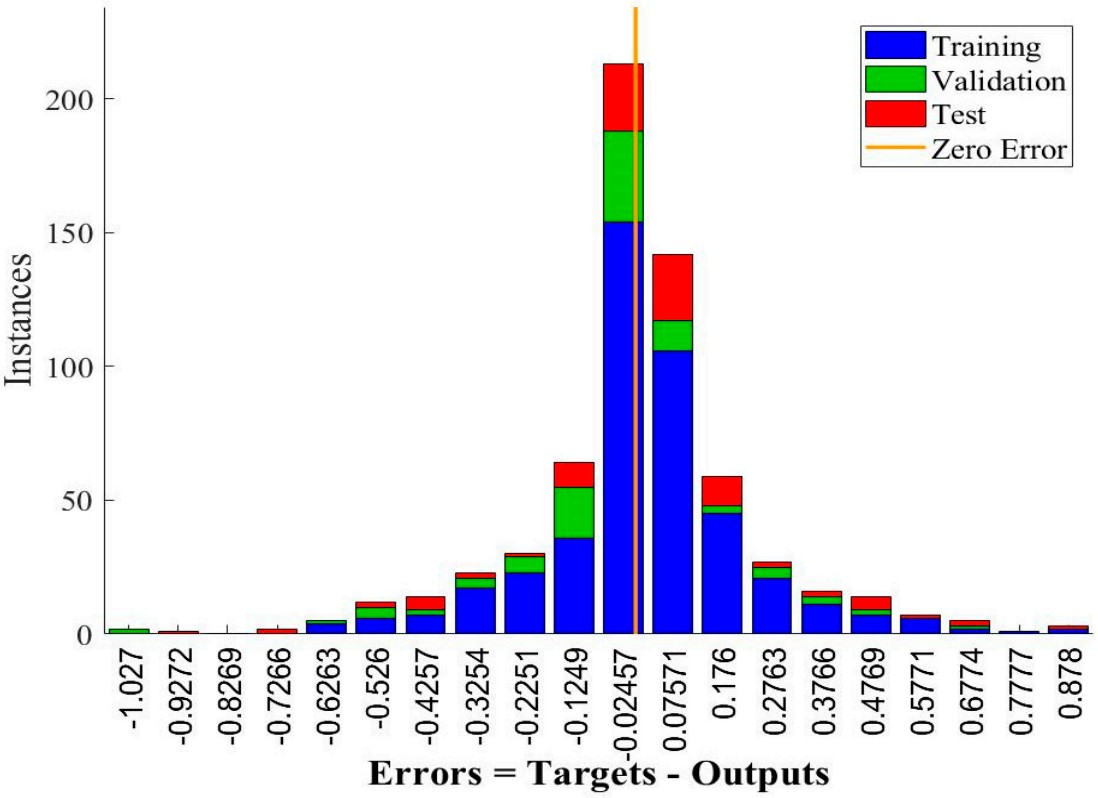

**Figure 9.** An Error histogram of ANN model.

*3.3. Models Predicted versus Actual Results*

For every one of the Fc, Fs, Ff and Ec parameters, Figure 10 displays an ANN predicted values and target values. The values predicted via the ANN model agree with the goal values perfectly, according to the results obtained in the Figures. The proposed ANN model can predict Fc, Fs, Ff and Ec values with excellent accuracy, as evidenced by the ideal fit between ANN estimations and target values. MoD values were produced for every data points and they are displayed in Figure 11, which expresses the proportional deviations between the values predicted from the proposed ANN model and the goal values. The data points reflecting the MoD values are placed near to the zero line error, when the figures presented for every of the Fc, Fs, Ff and Ec values are examined. The numbers, however, make it very evident that the MoD levels are quite low. The predicted mean MoD values for the Fc, Fs, Ff and Ec values were −0.28%, 0.14%, 0.87% and 1.17%, respectively.

The targeted values of the Fc, Fs, Ff and Ec values differ from the ANN outputs in Figure 12, and a more thorough investigation of the ANN model's predictive ability is planned. The graphs make it abundantly evident that the computed varaition values for every data points are small. The results of the MoD and difference values demonstrate that the constructed ANN model was developed to calculate Fc, Fs, Ff and Ec values with very low errors.

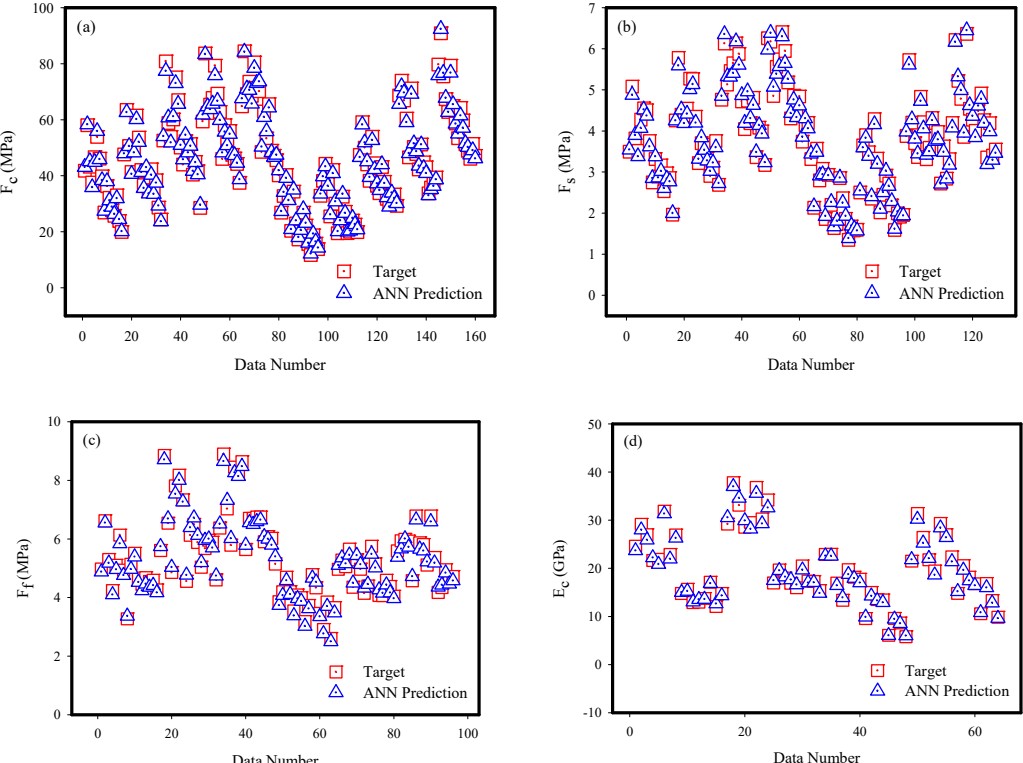

**Figure 10.** Predictions using ANN and targeting values considering data's number.

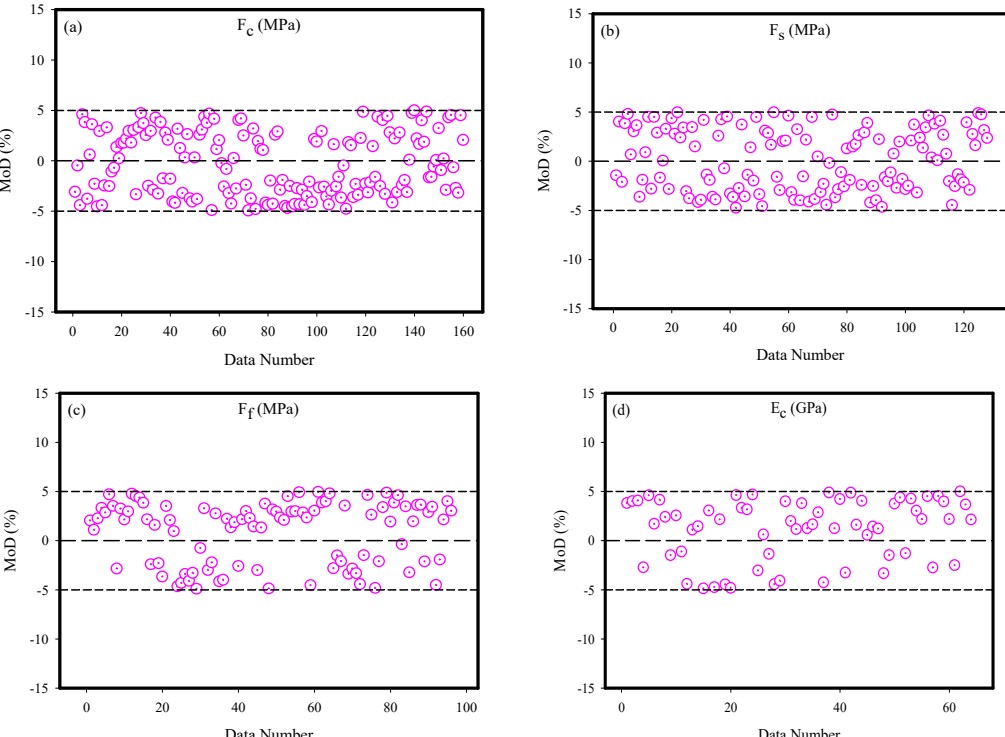

**Figure 11.** Values of MoD for each output considering data's number.

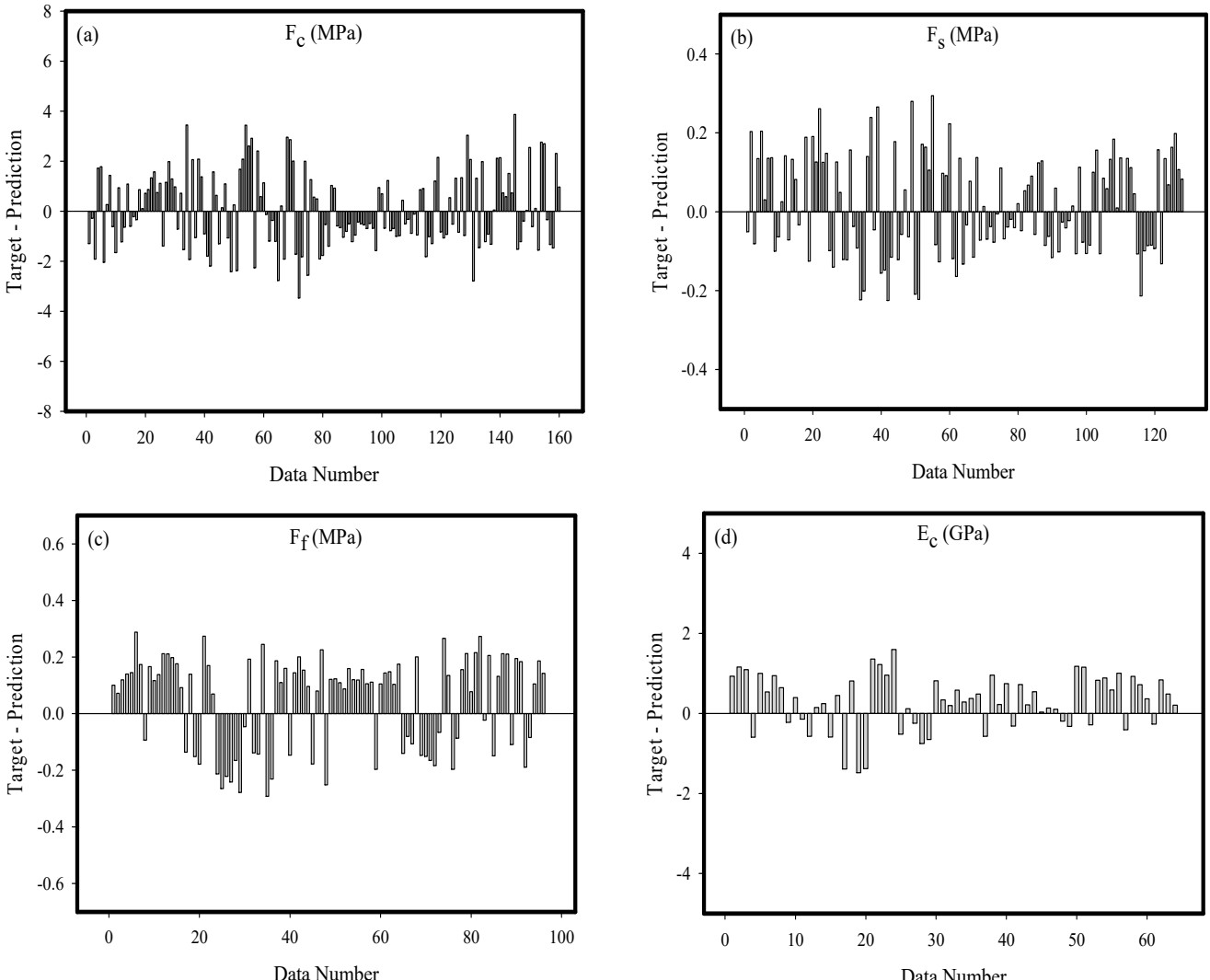

**Figure 12.** Various values considering data's number.

Figure 13 provides a clearer illustration of the agreement among the targeted values and ANN outputs. Every data point is illustrated to be close to the zero line error when looking at the locations of data points displayed for the Fc, Fs, Ff and Ec values. Additionally, it is noted that the data point fall within the 10 percent error limit. The MSE value for the created ANN model was calculated to be $6.45 \times 10^{-2}$, and the R value to be 0.99496. The created ANN model could be utilized in predicting Fc, Fs, Ff and Ec values having high accuracy based on the CR, NS, FA and *p* values, as can be observed from the perspective of all these results.

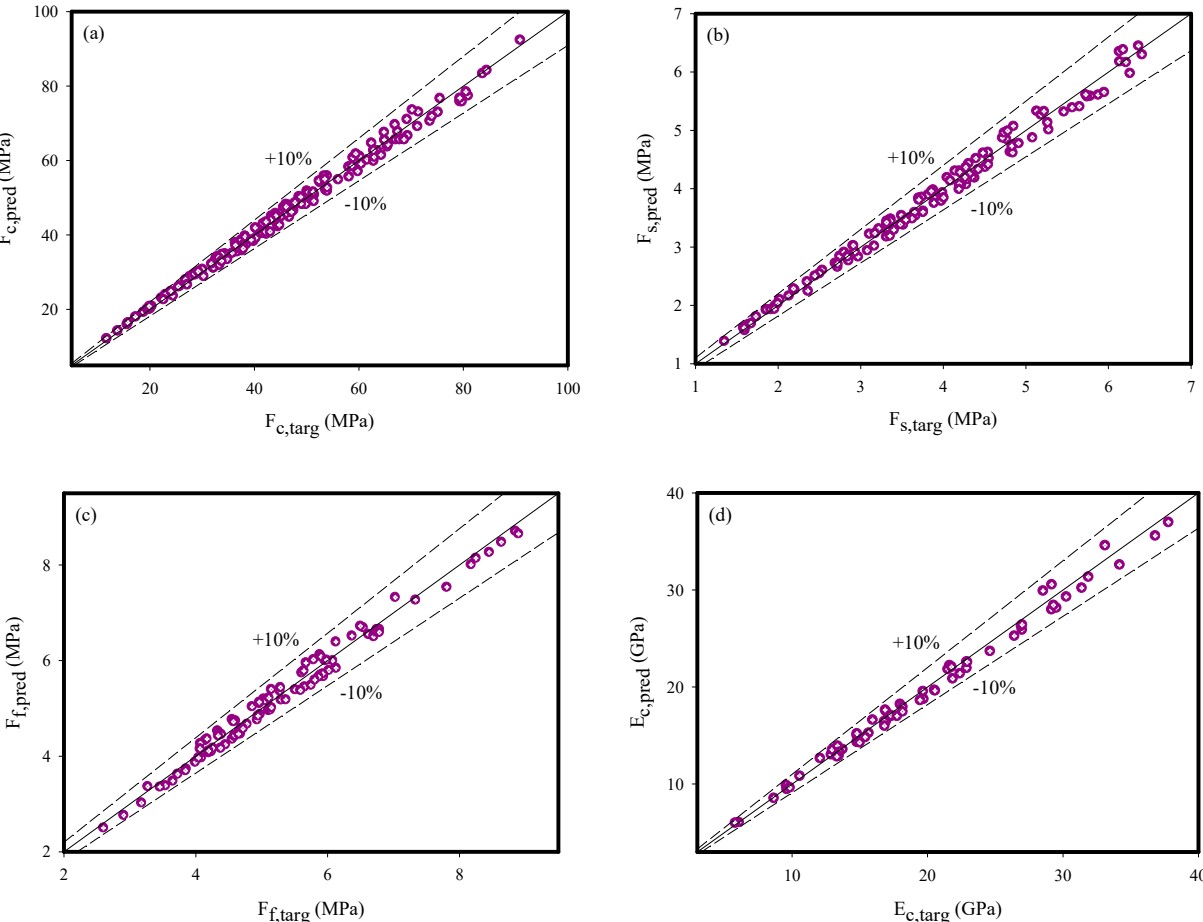

**Figure 13.** Target and prediction values for each output.

## 4. Conclusions

The ANN model is utilized to simulate the mechanical characteristics of concrete incorporating crumb rubber, nanosilica, and fly ash, including splitting tensile, compressive strength, elastic modulus, and flexural strength. The dataset for the modeling was obtained from the experimental results. The ANN model demonstrate more robust and accurate prediction skill in estimating the mechanical properties. Sensitivity analysis is utilized to optimize the ANN model's parameters, and compressive strength, a fundamental mechanical characteristic of concrete, is used to determine whether there is a linear or nonlinear relationships among an input parameters and targeted parameters. The outcome suggests that the most important factor in predicting strength is curing age. During the training phase, the proposed ANN model showed relatively low errors. The mean MoD values predicted values for Fc, Fs, Ff and Ec were −0.28%, 0.14%, 0.87% and 1.17%, respectively, which are near to the zero line. Overall, the ANN model predicted the strength with great accuracy. According to the experimental findings, fly ash and crumb rubber both reduced the mechanical strength of the concrete, however, the detrimental impact of the fly ash was only noticeable at young ages. Both the pozzolanic reactivity of fly ash and an impact of crumb rubber on mechanical characteristics of the concrete were partially alleviated by the addition of nanosilica.

Evaluating different properties such as durability generated from the modified concrete through adding admixtures in concrete such as, fibre, nanomaterial, ground glass fibre Therefore, it is recommended to capture and predict the overall concrete behavior considering these materials, and future work should focus on durability-related properties etc.

**Author Contributions:** Conceptualization, M.A. and Y.E.I.; methodology, M.A. and M.F.H.; software, A.B.Ç. and S.I.H.; validation, A.B.Ç. and S.I.H.; formal analysis, A.B.Ç. and S.I.H.; investigation, M.A. and M.F.H.; resources, Y.E.I.; data curation, M.A.; writing—original draft preparation, M.A., A.B.Ç. and S.I.H.; writing—review and editing, Y.E.I. and M.F.H.; visualization, M.A.; supervision, Y.E.I. All authors have read and agreed to the published version of the manuscript.

**Funding:** This research has no external funding. The authors acknowledge the support of Prince Sultan University in paying the article processing charges (APC) of this publication.

**Data Availability Statement:** Not applicable.

**Acknowledgments:** The authors wish to thank the Structures and Materials Research Laboratory of the College of Engineering Prince Sultan University Saudi Arabia for their viable support. Additionally, the authors would like to thank Prince sattam bin Abdulaziz University for supporting this work under project number PSAU/2023/R/1444.

**Conflicts of Interest:** The authors declare no conflict of interest.

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
