# Peer review of "Prediction of Mechanical Properties of Rubberized Concrete Incorporating Fly Ash and Nano Silica by Artificial Neural Network Technique"

_axioms, doi:10.3390/axioms12010081_

Round 1
Reviewer 1 Report
The authors conducted experimental program for rubberized concrete then utilized ANN method. Paper is good but needs to improved.
The authors more emphasize the importance of using recycled materials. Almost no significant sentecences and references were done.
Add a paragraph to discuss the use of recycled materials on
There are other materials to replace cement and aggregate such as waste glass, marble, bottom coal and etc. These should be indicated using followings: use of recycled coal bottom ash in reinforced concrete beams as replacement for aggregate; concrete containing waste glass as an environmentally friendly aggregate: a review on fresh and mechanical characteristics; mechanical behavior of crushed waste glass as replacement of aggregates;flexural behavior of reinforced concrete beams using waste marble powder towards application of sustainable concrete; Effects of Waste Powder, influence of replacing cement with waste glass on mechanical properties of concrete; Fine and Coarse Marble Aggregates on Concrete Compressive Strength; Investigation on improvement in shear performance of reinforced-concrete beams produced with recycled steel wires from waste tires; Performance evaluation of fiber-reinforced concretes produced with steel fibers extracted from waste tire; Improvement in Bending Performance of Reinforced Concrete Beams Produced with Waste Lathe Scraps
Novelty of paper is not clear. What is the difference with other studies
The size of crumb rubbers is higher than fine aggregate. Why the authors did not use same size materials?
The authors can include followings to emphasize the importance of ANN: Prediction Models for Marshall Mix Parameters Using Bio-inspired Genetic Programming and Deep Machine Learning Approaches: A Comparative Study; Free vibration analysis of open-cell FG porous beams: analytical, numerical and ANN approaches
Include both Vf and Wf ratio of crumbed rubber.
Why did you use only your data for ANN modeling. There are lots of studies on this subject. You can include them to increase accuracy
Add photo of rubber
Add photo of test setup
Concluison section is only based on ANN. Improve this section
Include future needs and recommendations
Reviewer 2 Report
The paper can be published after the corrections have been made. Please send the paper again after the authors make corrections.

Reviewer 3 Report
The submitted article, “Prediction of mechanical Properties of rubberized concrete incorporating fly ash and nano silica by artificial neural network technique” is interesting, original and within the scope of the journal but some changes should be addressed:
1. Please update the state of the art by giving some examples of admixtures materials properties (e.g. surface, thermal stability) as you mentioned in the text from line 42 to 45 (please see https://doi.org/10.37358/RC.19.5.7161) in order to understand how the material could influence the properties of final product.
2. At line 113 the authors used commercial nanosilica, please mention the type, manufacturer and the country. Also, please give information regarding the fly ash origin.
3. Please use the same font and size for text in figures.
4. Try to eliminate the gridlines from figure 1.
5. I recommend to give informations regarding the reproducibility and the repeatability of the experiments.
6. Try to reformulate the results section in order to have results and discussion.
7. Please rewrite the references according to journal instructions.
Round 2
Reviewer 1 Report
The paper can be accepted in this current form